



# Tracer-based investigation of organic aerosols in marine atmospheres from marginal seas of China to the northwest Pacific Ocean

Tianfeng Guo[1], Zhigang Guo[1], Juntao Wang[2], Jialiang Feng[3*], Huiwang Gao[2,4], Xiaohong Yao[2,4*]

[1] Shanghai Key Laboratory of Atmospheric Particle Pollution and Prevention, Department of Environmental Science and Engineering, Fudan University, Shanghai 200433, China;

[2] Lab of Marine Environmental Science and Ecology, Ministry of Education, Ocean University of China, Qingdao 266100, China

[3] School of Environmental and Chemical Engineering, Shanghai University, Shanghai 200444, China

[4] Pilot National Laboratory for Marine Science and Technology (Qingdao), Qingdao, China

*Correspondence to*: Xiaohong Yao (xhyao@ouc.edu.cn); Jialiang Feng (fengjialiang@shu.edu.cn)

**Abstract.** We investigated the geographic distributions of organic tracers in total suspended particles over marginal seas of China, including the Yellow and Bohai seas (YBS) and the South China Sea (SCS), and the northwest Pacific Ocean (NWPO) in spring, when Asian outflows strongly affect downwind marine atmospheres. The comparison of levoglucosan observed in this study with values from the literature implied that the contribution from biomass burning emissions to marine aerosols over the NWPO may have increased largely over the last decades. The increase led to the mean value of levoglucosan ($8.2\pm14$ ng m$^{-3}$) observed over the NWPO closer to that over the SCS and almost half of that over the YBS. Small geographic differences in monoterpene-derived and sesquiterpene-derived secondary organic tracer concentrations were obtained among the three atmospheres, although the causes may differ. By contrast, a large difference in isoprene-derived secondary organic tracer concentrations was observed among the three atmospheres, with the sum of tracer concentrations over the SCS ($45\pm54$ ng/m$^3$) several times and approximately one order of magnitude greater than that over the YBS ($15\pm16$ ng/m$^3$) and the NWPO ($2.3\pm1.6$ ng/m$^3$), respectively. The geographic distribution of aromatic-derived secondary organic tracers was similar to that of isoprene-derived secondary organic tracers, with a slightly narrower difference, i.e., $1.8\pm1.7$ ng/m$^3$, $1.1\pm1.4$ ng/m$^3$ and $0.3\pm0.5$ ng/m$^3$ over the SCS, the YBS and the NWPO, respectively. We discuss the causes of the distinctive geographic distributions of these tracers and present the tracer-based estimation of organic carbon.

## 1 Introduction

Aerosols that emanate from biomass burning (BB) consist primarily of carbonaceous components and inorganic salts, which can affect the climate directly by absorbing solar radiation or indirectly by acting as either cloud condensation nuclei (CCN) or ice nuclei (IN) (Bougiatioti et al., 2016; Chen et al., 2017; Hsiao et al., 2016). High BB aerosol emissions zones include boreal forests (e.g., in Eurasia and North America), tropical forests (e.g., in southeast Asia and the tropical Americas), and agriculture areas where crop residuals are burned (e.g., in developing countries such as China and India, etc.) (van der Werf et al., 2006). BB aerosols can undergo long-range transport in the atmosphere, which can carry them from the continents to the oceans (Ding et al., 2013; Fu et al., 2011; Kanakidou et al., 2005). For example, BB aerosols from boreal forest wildfires in Russia



and China reportedly made an appreciable contribution to atmospheric particle loads observed over the Arctic
Ocean and northwestern Pacific Ocean (NWPO) based on specific tracers of BB (Ding et al., 2013). Although
open wildfires from forests occur sporadically in terms of strength and occurrence frequency, global warming
could be conducive to vegetation fires (Running, 2006) and thus increase emissions of BB aerosols. In this
century, nine years were among the ten hottest global years on record, with 2014–2018 being ranked as the top
five hottest years (https://www.climatecentral.org/gallery/graphics/the-10-hottest-global-years-on-record). The
question is automatically raised: how do BB aerosols in the marine atmosphere in the hottest global years
change against those observations previously reported?
In addition to BB aerosols, secondary oxidation of biogenic volatile organic compounds (BVOCs) and
anthropogenic VOCs (AVOCs) also contribute to the particulate carbonaceous components of marine
atmospheres (Kanakidou et al., 2005). Many field and modeling studies have proposed that secondary organic
aerosols (SOAs) arising from the oxidation of phytoplankton-derived isoprene may affect the chemical
composition of marine atmospheric aerosols and consequently impact CCN loading and cloud droplet number
concentrations (Ekström et al., 2009; Meskhidze and Nenes, 2006; Claeys et al., 2004). Several modeling
studies have shown that the NWPO may experience the greatest increases in sea surface temperature and $CO_2$
input under a future warming climate in the future (John et al., 2015; Lauvset et al., 2017). The Kuroshio
Extension current system leads the NWPO to be an active subtropical cyclone basin, promoting biogenic
activities (Hu et al., 2018). From the perspective of global change, it is a long-term need to study the dynamic
changes in atmospheric aerosols derived from marine sources over the NWPO and adjacent marginal seas of
China, as well as their potential effects on climate.
Not limited by phytoplankton-derived isoprene, BVOCs emitted from continental ecosystems and their
oxidation products can also affect the atmosphere in remote marine areas through long-range transport (Hu et al.,
2013a; Ding et al., 2013; Kang et al., 2018; Fu et al., 2011; Kawamura et al., 2017). BVOCs consist primarily of
isoprene, monoterpenes, sesquiterpenes, and their oxygenated hydrocarbons such as alcohols, aldehydes, and
ketones (Guenther et al., 2006; Ehn et al., 2014) and account for the majority of the global VOC inventory (Zhu
et al., 2016a, b; Heald et al., 2008). However, emissions fluxes and oxidation processes of BVOCs show great
variation, depending on global warming and other factors such as regional landscape, other pollutants in the
ambient air etc. (Ait-Helal et al., 2014; Hu and Yu, 2013; Peñuelas and Staudt, 2010). Unlike a potential
increase in BVOC-derived organics aerosols in marine atmospheres under global warming, anthropogenic
VOCs and carbonaceous particles over the continents have been decreased because of effective mitigation of air
pollutants in the last decades (Sharma, 2004; Murphy et al., 2011; Zhang et al., 2012). In the northern
hemisphere, marine atmospheres are also usually affected by anthropogenic pollutants to some extent, most of
which are derived from long-range transport from continents (Kang et al., 2019; Bao et al., 2018; Zhang et al.,
2017). The revere trends in BVOC and anthropogenic VOC would change the composition, sources of
carbonaceous particles in marine atmospheres. Update observations are thereby needed to reveal the change and
service the future study of the impacts.
In this study, we analyzed the concentrations of some typical organic tracers in aerosol samples obtained from
three cruise campaigns from the marginal seas of China, including in the South China Sea (SCS) in 2017,
Yellow Sea and Bohai Sea (YBS), to the NWPO in 2014, both in springtime. We investigated the influences of
BB aerosols from continents over three marine atmospheres, quantified the contributions of various precursors
to the observed SOA in marine atmospheres using organic tracers established in the literature, and explored the
formation pathways of SOA from their precursors during long-range transport in these hottest global years.
Particularly, we conducted a comprehensive comparison of this observation with those reported in literature in
terms of long-term variations and geographic distributions of these tracers, etc.



## 2 Materials and Methods

Total suspended particulate (TSP) samples were collected over the NWPO from 19 March to 21 April 2014, over the YBS from 30 April to 17 May 2014, and over the SCS from 29 March to 4 May 2017. All samples were collected on the upper deck of the R/V Dong Fang Hong II, which sits ~8 m above the sea surface. To avoid contamination from the ship's exhaust, samples were collected only when the ship was sailing, and the wind direction ranged from –90° to 90° relative to the bow. TSP samples were collected on quartz fiber filters (Whatman QM-A) that had been pre-baked for 4 h at 500°C prior to sampling using a high-volume sampler (KC-1000, Qingdao Laoshan Electric Inc., China). The sampling duration was 15–20 h at a flow rate of ~1000 L /min. After sampling, the sample filters were wrapped in baked aluminum foil and sealed in polyethylene bags, then stored at -20°C and transported to the laboratory. Field blanks were collected during each sampling period. However, one sampler was out of service during the cruise on the SCS. As a compromise, cellulose filters (Whatman 41) previously intended for elemental analyses were used for analyses of the organic tracers in TSP.

The method for determining the concentrations of tracers was adapted from Kleindienst et al. (2007) and Feng et al. (2013). Briefly, 20 mL dichloromethane/methanol (1:1, v/v) was used for ultrasonic extraction of 40 cm$^2$ of each filter at room temperature three times. The combined extracts were filtered, dried under a gentle stream of ultrapure nitrogen, and then derivatized with 100 µL N,O-bis-(trimethylsilyl)-trifluoroacetamide (BSTFA, containing 1% trimethylchlorosilane as a catalyst) and 20 µL pyridine at 75°C for 45 min. Gas chromatography mass spectrometry (GC-MS) analyses were conducted with an Agilent 6890 GC/5975 MSD. Prior to solvent extraction, methyl-β-D-xylanopyranoside (MXP) was spiked into the samples as an internal/recovery standard. Hexamethylbenzene was added prior to injection as an internal standard to check the recovery of the surrogates.

Like those reported by Feng et al. (2013), the primary organic tracers analyzed in this study included levoglucosan (LEVO), mannosan, and galactosan. Four types of secondary organic tracers were used: isoprene-derived secondary organic tracers (SOA$_I$) including 2-methylglyceric acid (2-MGA), C5-alkene triols (cis-2-methyl-1,3,4-trihydroxy-1-butene, 3-methyl-2,3,4-trihydroxy-1-butene and trans-2-methyl-1,3,4-trihydroxy-1-butene), and MTLs (2-methylthreitol and 2-methylerythritol); monoterpene-derived secondary organic tracers (SOA$_M$) including 3-hydroxyglutaric acid (HGA), 3-hydroxy-4,4-dimethylglutaric acid (HDMGA), and 3-methyl-1,2,3-butanetricarboxylic acid (MTBCA); the sesquiterpene-derived secondary organic tracer (SOA$_S$) β-caryophyllinic acid; and the aromatic (toluene)-derived secondary organic tracer (SOA$_A$) 2,3-dihydroxy-4-oxopentanoic acid (DHOPA). LEVO was quantified based on authentic standards in this study. While the SOA tracers without available commercial standards were quantified using methyl-β-D-xylanopyranoside (MXP) as a surrogate. To reduce the uncertainty of quantification, relative response factors of the target tracers to MXP were estimated by comparing the area ratio of typical target ions to MXP to that of total ions in selected samples that showed high concentrations of the target tracers (Feng et al., 2013).

Field blanks and laboratory blanks (run every 10 samples) were extracted and analyzed in the same manner as the ambient samples. Target compounds were nearly always below the detection limit in field and laboratory blanks. Recoveries of the surrogate (MXP) were in the range of 70–110%. The reported results were corrected for recovery, assuming that the target compounds had the same recovery as the surrogate. Duplicate analyses indicated that the deviation was less than 15%.

However, the substitution of cellulose filters (Whatman 41) during the cruise on the SCS led to increased field blank values for some tracers. The tracer concentrations in those samples were, however, over three times higher than the field blank values, except for those of mannosan and galactosan. Data for mannosan and galactosan were thus not available, nor were the total organic carbon concentrations, for samples collected during the cruise





on the SCS.
The concentrations of organic carbon (OC) and element carbon (EC) in each sample were analyzed with a DRI
2001A thermal/optical carbon analyzer (Atmoslytic Inc., Calabasas, CA, USA) using the IMPROVE
temperature program (Wang et al., 2015).
3. Results and Discussion
3.1 Spatiotemporal distributions of LEVO
Levoglucosan, mannosan, and galactosan produced by the pyrolysis of cellulose and hemicellulose have been
widely used as organic tracers of BB aerosols in ambient air (Ding et al., 2013; Fu et al., 2011; Feng et al.,
2013). The mean levels of LEVO in TSP collected during the cruises on the NWPO and the SCS were
comparable, at 8.2 ng/m$^3$ and 9.6 ng/m$^3$, respectively (Figure S1, Table 1). They were almost half of the mean
value of 21 ng/m$^3$ during the cruise on the YBS, where high concentrations of BB aerosols have been observed
in continental atmospheres upwind of the YBS mainly from wildfires and the burning of burning crop residue,
wildfire, etc. (Yang et al., 2014; Feng et al., 2012; Feng et al., 2013). Unlike the smaller difference among the
means values, the concentration of LEVO fluctuated greatly among TSP samples in each oceanic zone, ranging
from 0.5 to 65 ng/m$^3$ over the NWPO, from 1.0 to 30 ng/m$^3$ over the SCS and from 2.5 to 42 ng/m$^3$ over the
YBS (Fig. S1). High spatiotemporal variation in LEVO in TSP has also been observed in literature, with
concentrations of LEVO fluctuating around 0.2–41 ng/m$^3$ during Arctic to Antarctic cruises from July to
September 2008 and from November 2009 to April 2010 (Hu et al., 2013b). Hu et al. (2013b) also reported the
highest LEVO concentrations occurring at mid-latitudes (30°–60° N and S) and the lowest at Antarctic and
equatorial latitudes over the several months of sampling. This distinctive geographical distribution was not
observed in the present study, as there were no significant differences in LEVO in TSP between the SCS and
NWPO (P > 0.05).
Narrow spatiotemporal variation in LEVO in TSP has been reported during summer sampling over the North
Pacific Ocean and the Arctic in 2003, with maximum and mean values as low as 2.1 ng/m$^3$ and 0.5 ng/m$^3$,
respectively (Ding et al., 2013). A lower mean value of LEVO of 1.0 ng/m$^3$ has also been reported in the spring
over the island of Chichi-jima from 2001 to 2004 (Mochida et al., 2010), while the levels increased to 3.1 ± 3.7
ng/ m$^3$ in TSP collected on the island of Okinawa in 2009–2012 (Zhu et al., 2015). Using these previous
observations as a reference, our observations suggested that the contribution of BB aerosols to particle loading
over the NWPO may have increased continuously and largely over the last decades.
Due to the lack of BB sources in oceans, large spatiotemporal variation in the concentrations of LEVO in the
marine atmosphere may be related to the long-range transport of atmospheric particles from continents. Thus, 72
h back trajectories of air masses at a height of 1000 m during our sampling periods (Figs. 1, 2) were calculated
using the HYSPLIT model (https://ready.arl.noaa.gov/HYSPLIT). Based on the calculated back trajectories, TSP
samples could be classified into two categories with Category 1 representing continent-derived aerosol samples
and Category 2 being ocean-derived aerosol samples. All 12 samples collected over the YBS fell into Category 1
(Fig. 2). Half (11/19) of the samples collected over the NWPO were classified into Category 1 (Fig. 1). A
significant difference (p < 0.05) was obtained between the concentrations of LEVO in Category 1 (13±18 ng/m$^3$)
and Category 2 (2.0 ±1.8 ng/m$^3$) samples over the NWPO. The values in Category 2 were closer to the
springtime observations reported by Mochida et al. (2010) and Zhu et al. (2015) as well as the summer
observations reported by Ding et al. (2013), reflecting the marine background value less affected by continental
air masses. On the other hand, the much higher values in Category 1 than Category 2 further indicated a large



increase in contribution of BB aerosols being transported from the continents to the remote marine atmosphere
in 2014.
On 11 April 2014 over the NWPO, an episode of high LEVO concentration of 65 ng/m$^3$ occurred (Fig. 1). Like
LEVO, the concentrations of galactosan and mannosan in the sample were also the highest among all samples
collected over the NWPO. This sample was collected in the oceanic zone, approximately 500 km from the
continent of Japan. A combination of air mass back traceries and NASA's FIRMS Fire Map indicated strong BB
aerosol emissions from intense fire events in Siberia, followed by long-range transport with the westerly wind as
the major contributors to this anomaly (Fig. 1). A similar episodic concentration of LEVO of 27 ng/m$^3$ in TSP
was observed once previously over the NWPO during a circumnavigation cruise (Fu et al., 2011). By combining
satellite data with other observations, many studies in literature have found that BB aerosols from major forest
fires and smoke events in Siberia could be transported downwind to remote marine regions not only in spring,
but also in summer (Generoso et al., 2007; Ding et al., 2013; Huang et al., 2009). In a few cases, BB aerosols
have been reported to have reached as far as the adjacent Arctic region (Warneke et al., 2010; Generoso et al.,
2007). Van der Werf et al. (2006) estimated the emissions of BB aerosols from Eurasia to be much larger than
those from North America. Thus, it is not surprised that the concentrations of LEVO over the NWPO were much
higher than those over the eastern North Pacific and western North Atlantic at similar latitudes (Hu et al.,
2013b).
In addition, both galactosan and mannosan showed strong linear correlations with LEVO ($R^2 = 0.98$, $p < 0.05$)
in TSP collected over the NWPO and YBS in this study. These strong correlations indicate that the three tracers
were probably derived from the same BB sources. Previous studies have reported LEVO/mannosan (L/M) ratios
of 3–10, 15–25, and 25–40 from softwood, hardwood, and crop-residue burning, respectively (Kang et al., 2018;
Zhu et al., 2015). The calculated L/M ratios in TSP collected over the NWPO were 19±4 in this study, which
implies dominant contributions from herbaceous plants and hardwood. The calculated L/M ratios in TSP
collected over the YBS were 14±11, indicating mixed sources.
In all, 5 of 13 samples collected over the SCS were classified into Category 1, with air masses identified as
originating from either the continental areas of South China or the Philippines (Fig. 2). The concentration of
LEVO fluctuated around 17±12 ng/m$^3$ in Category 1 but decreased to 3.6±3.4 ng/m$^3$ in Category 2. However, no
significant differences were found between categories due to the large variation in LEVO concentration among
the limited samples in Category 1 ($p > 0.05$). Forest fires occur accidentally, leading to the large variation in
LEVO in Category 1. Southern Asia has been reported to be one of the greatest emissions sources of BB
aerosols worldwide (van der Werf et al., 2006), which likely led to the higher mean value of LEVO in Category
1. However, the LEVO level observed over the SCS in Category 2 was closer to that reported from low-latitude
regions (2.7±1.1 ng/m$^3$, Table 1) collected during a global circumnavigation cruise (Hu et al., 2013b). Hu et al.
(2013b) argued that their low observed concentrations may have been associated with intense wet deposition,
degradation as well as intensive moist convection that occurred in the tropical region during their summer cruise.
Unfortunately, no previous observations of LEVO in spring can allow us analyzing the long-term variation in
contribution of BB aerosols therein. However, this observation can be used for future comparison.



**Figure 1. Spatial distribution of LEVO over the NWPO (2014) and BTs corresponding to the samples; the samples with the backward trajectories (red lines) indicate land-influenced aerosols (a, Category 1) and the blue line denotes ocean-influenced aerosols collected (b, Category 2). The red dots represent the locations of fires from Fire Information for Resource Management System (FIRMS,**





https://firms.modaps.eosdis.nasa.gov/). **The base map was from Resource and Environment Data Cloud**
**Platform, DOI: 10.12078/2018110201.**

**Figure 2. Spatial distribution of LEVO over the YBS (a, 2014), and NWPO (b, 2017), detailed information**

**described in Figure 1.**




### 3.2 Spatiotemporal distributions of SOA$_I$ tracers

SOA$_I$ tracers were detected during all three cruises. The sum of SOA$_I$ tracers showed a decreasing trend of up to
approximately one order of magnitude from marginal seas to the open ocean, i.e., $45 \pm 54$ ng/m$^3$ in TSP over the
SCS, $15 \pm 16$ ng/m$^3$ over the YBS and $2.3 \pm 1.6$ ng/m$^3$ over the NWPO (Fig. S1). The highest sum value of SOA$_I$
tracers over the SCS was 176 ng/m$^3$, indicating strong photochemical formation of SOA from biogenic volatile
organics (Fig. 3). The geographical distribution of SOA$_I$ tracers in this study was generally consistent with those
reported by Hu et al. (2013a), with higher concentrations of these tracers in atmospheric particles collected from
low-latitude oceanic zones (30° S–30° N) due to large emissions from tropical forests and strong photochemical
reactions. Their reported average contents of SOA$_I$ tracers in low-latitude oceanic zones fluctuated around
$9.2\pm6.7$ ng/m$^3$, much lower than those measured in this study.
When the sum of SOA$_I$ tracers in each sample was examined separately according to the air mass source, a
significant difference was found over the SCS between Category 1 ($85\pm66$ ng/m$^3$) and Category 2 ($19\pm22$
ng/m$^3$), with significance at $p < 0.01$. The tracer values were $2.7\pm1.8$ ng/m$^3$ in Category 1 and $1.7\pm1.0$ ng/m$^3$ in
Category 2 over the NWPO, where no significant difference between the two categories was found ($p > 0.05$).
Supposed that concentrations of the tracers in Category 2 were completely contributed by marine sources, it can
be inferred that SOA$_I$ carried by continental air masses increased sharply over the SCS. However, it was not the
case over the NWPO. Because all samples over the YBS fell into Category 1, this comparison could not be
made for the YBS.



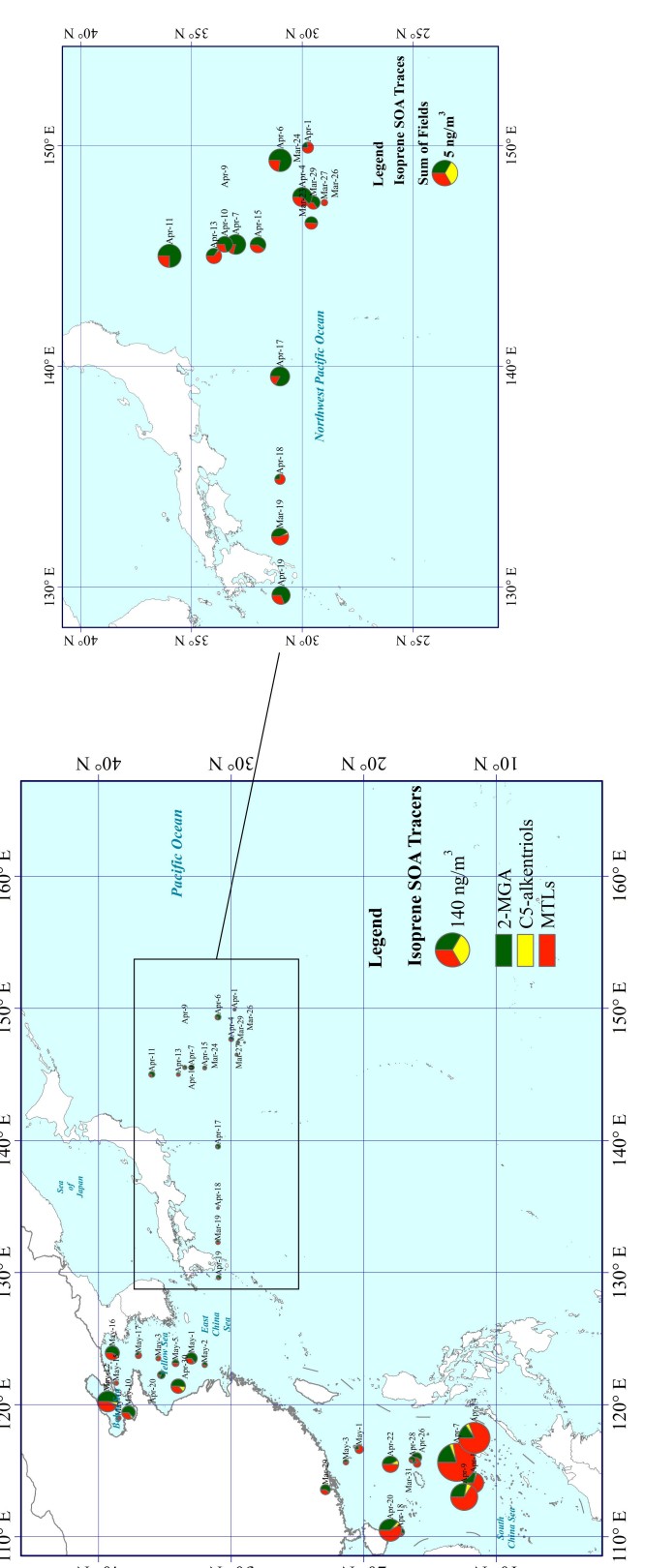

**Figure 3. Spatial distribution of SOA$_I$ tracer compounds over three marine regions, ECS and NWPO in 2014, SCS in 2017. The area of the pie indicates the concentration of total SOA$_I$ tracers. The base map was from Resource and Environment Data Cloud Platform, DOI: 10.12078/2018110201.**





**3.3 Spatiotemporal distributions of SOA$_M$, SOA$_s$ tracers**
The sum of SOA$_M$ tracers including HGA, HD-MGA, and MBTCA was greatest over the SCS region ($3.5\pm6.0$
ng/m$^3$), where the concentration was approximately double that over the YBS ($1.6\pm2.0$ ng/m$^3$) and NWPO
regions ($1.6\pm2.7$ ng/m$^3$) (Fig. S1), but no significant differences were identified between any two campaigns.
The concentrations of SOA$_M$ tracers were almost one magnitude lower than those of SOA$_I$ tracers. Due to the
unique contribution of terpene-derived SOA to nucleation and growth of newly formed particles in the
atmospheres (Gordon et al., 2017; Zhu et al., 2019; Ehn et al., 2014), the SOA$_M$ may primarily cause indirect
climate effects rather than direct effects of aerosols in the marine atmosphere. The difference in mean SOA$_M$
concentration between the SCS and NWPO narrowed to a factor of two, in contrast to the differences of
approximately one order of magnitude in mean SOA$_I$ between the two types of atmospheres. The precursors of
SOA$_M$ tracers derive mainly from coniferous forests (Duhl et al., 2008) and the decreasing proportion of
coniferous forests in subtropical and tropical regions may partially explain the smaller spatial difference in
SOA$_M$ tracers over the SCS compared to the YBS and NWPO. However, the comparable SOA$_M$ levels over the
YBS and NWPO have not yet been explained.
Only three SOA$_M$ tracers were measured in this study, but other SOA$_M$ tracers have been measured and reported
in marine atmospheres (Kang et al., 2018; Fu et al., 2011). In order to compare our results with the total amount
of SOA$_M$ tracers in the literature, the total amounts measured in this study were multiplied by a factor of 3.1
(described in supporting information Sect. S1, Fig. S4) according to the chamber results obtained by Kleindienst
et al. (2007). The adjusted values over the SCS were closer to the mean of 11.6 ng/m$^3$ observed over the ECS
(Kang et al., 2018) and the lower values of 9.80–49.0 ng/m$^3$ observed among 12 continental sites in China (Ding
et al., 2016). The adjusted total amounts of SOA$_M$ over the NWPO and YBS were comparable to previous
observations of $3.0\pm5.0$ ng/m$^3$ collected from the Arctic to Antarctic in 2008-2010 (Hu et al., 2013a), but much
higher than observations of $63\pm49$ pg/m$^3$ over the North Pacific and Arctic in 2003 (Ding et al., 2013). This may
also imply a substantial increase in SOA$_M$ in the last decades, although more investigations are needed to
confirm.
β-Caryophyllene is a major sesquiterpene emitted from plants such as Scots pine and European birch (Duhl et al.,
2008; Tarvainen et al., 2005). β-Caryophyllinic acid is formed through the ozonolysis or photo-oxidation of
β-caryophyllene. The highest levels of β-caryophyllinic acid were observed over the YBS ($0.13\pm0.03$ ng/m$^3$),
followed by the SCS ($0.08\pm0.11$ ng/m$^3$) and NWPO ($0.05\pm0.09$ ng/m$^3$) (Fig. S1). The spatial distribution of
β-caryophyllinic acid clearly did not follow the general trend of biogenic SOA, with the highest values over the
SCS followed by the YBS. Compared to values from the literature, our results are much higher than those over
the North Pacific and Arctic Oceans ($2.4\pm5.4$ pg/m$^3$) (Ding et al., 2013) but much lower than observations over
the East China Sea reported by Kang et al. (2018), where β-caryophyllinic acid was reported to be in the range
of 0.16–17.2 ng/m$^3$ with a mean of 2.9 ng/m$^3$. The large differences in β-caryophyllinic acid content observed in
various campaigns remains unexplained.
**3.4 Spatiotemporal distributions of SOA$_A$ tracers**
When the concentrations of DHOPA in TSP were examined, the highest concentrations occurred over the SCS
($1.8\pm1.7$ ng/m$^3$), followed by the YBS ($1.1\pm1.4$ ng/m$^3$), and the lowest values were recorded in the NWPO
region ($0.3\pm0.5$ ng/m$^3$) (Fig. S1). The extent of the DHOPA decrease from the SCS to the NWPO was
approximately three times less than that of SOA$_I$ tracers but approximately three times larger than that of SOA$_M$





tracers. Ding et al. (2017) reported annual averages of DHOPA among various sites in China, which ranged from
1.2 to 8.8 ng/m$^3$. The concentrations of DHOPA observed over the SCS and the YBS were similar to the lower
values observed in upwind continental atmospheres.
Formation of DHOPA depends on the molecular structures of aromatics, as well as concentrations of free
radicals and oxidants, etc. (Li et al., 2016; Henze et al., 2008). The mean value of DHOPA in Category 1
(0.43±0.65 ng/m$^3$) was nearly twice that in Category 2 (0.20±0.31 ng/m$^3$) over the NWPO (p > 0.05). With two
samples with high DHOPA (1.2, 2.1 ng/m$^3$) in Category 1 to be excluded, the recalculated average DHOPA
decrease down to 0.17±0.21 ng/m$^3$. The continent-derived DHOPA seemingly yielded a minor contribution to
the observed values over the NWPO, except during strong long-range transport episodes. Similarly, the mean
values of DHOPA were same in Category 1 (1.8±2.1 ng/m$^3$) and Category 2 (1.8±1.5 ng/m$^3$) samples collected
over the SCS and no significant differences were observed between two categories. Much stronger UV radiation
occurs over the SCS than the YBS, which may contribute to the elevated DHOPA level over the SCS. Aside
from continent-derived precursors, oil exploration and heavy marine traffic over the SCS are also potential
contributors to the higher DHOPA levels therein, and this link requires further investigation. Previous field
observations in China have demonstrated that biofuel or biomass combustion emissions act as important sources
of aromatics in the atmosphere (Zhang et al., 2016), as evidenced by the association between the nationwide
increase in DHOPA during the cold period and the enhancement of BB emissions (Ding et al., 2017). In this
study, no linear correlation was obtained between DHOPA and LEVO in samples collected over the SCS and
other two campaigns, leaving emissions other than BB emissions as the major precursors for DHOPA in these
marine atmospheres (Li et al., 2013).
3.5 Causes for high photochemical yields of SOA$_I$ over the SCS
Because higher concentrations of SOA$_I$ were observed in TSP samples collected over the SCS, the composition
of SOA$_I$ tracers was further investigated in terms of their formation pathways and sources. Based on the results
of chamber experiments, Surratt et al. (2010) proposed different formation mechanisms for 2-MGA and MTLs.
2-MGA is a C4-dihydroxycarboxylic acid, which forms through a high-NO$_x$ pathway. MTLs and C5-alkene
triols are mainly products of the photooxidation of epoxydiols of isoprene under low-NO$_x$ conditions.
MTLs acted as the dominant compounds among SOA$_I$ tracers in most TSP samples collected over the SCS, with
concentrations of 31±42 ng/m$^3$ (Fig. 3). The ratio of 2-MGA/MTLs ranged from 0.2 to 3.1, with a median value
of 0.6. The ratio exceeded the unity in only 4 of 13 samples. This result allowed us to infer that the observed
SOA$_I$    tracers    were    generated    mainly    under    low-NO$_x$    conditions.    Although    the    concentration    of
2-methylerythritol was nearly double that of 2-methylthreitol, they were highly correlated (R$^2$ = 0.99, p < 0.05)
because of their shared formation pathway. Satellite data showed that the NO$_2$ levels in South China and the
Philippines were low, except in a few hotspots (Fig. S2). Such low-NOx conditions favor the formation of
MTLs rather than 2-MGA over the tropical SCS. The isoprene emitted from plants growing on oceanic islands
may also undergo chemical conversion to SOA under low-NOx conditions, and low-NOx conditions are always
expected in remote marine atmospheres (Davis et al., 2001).
In general, zonally and monthly averaged OH concentrations around 15°N are ~50% greater than those around
35 ºN (Bahm and Khalil, 2004). Thus, enhanced formation of MTLs is theoretically expected under the strong
UV radiation of tropical regions. However, no significant correlation between the concentrations of MTLs and
UV radiation was obtained over the SCS (data not shown) possibly due to the influences of various air masses.
A field study showed that MTL yields were positively correlated with ambient temperature in continental
atmospheres (Ding et al., 2011). 2-MGA yields, in contrast, showed no significant correlation with ambient





temperature in this study. Moreover, lower relative humidity may enhance the formation of 2-MGA in the
particulate phase but not for MTLs (Zhang et al., 2011). Variation in ambient temperature and relative humidity
may complicate the relationship between the concentrations of $SOA_I$ tracers and UV radiation over the SCS.
In addition, the MTLs concentration in Category 1 ($62\pm55$ ng/m$^3$) was larger than that for Category 2 ($11\pm14$
ng/m$^3$). The more abundant MTLs associated with Category 1 was most likely related to long-range transport of
these chemicals from upwind continental areas, the oxidation of continental precursors in the marine atmosphere,
or both. Large emissions of isoprene were expected from tropical forests upwind of the SCS due to the high
vegetation coverage and high ambient temperature of such areas (Ding et al., 2011; Rinne et al., 2002). Global
estimates show tropical trees to be responsible for ~80% of terpenoid emissions and ~50% of other VOC
emissions (Guenther et al., 2012).
In a clean marine atmosphere, phytoplankton is the sole source of isoprene emissions over the oceans (Bonsang
et al., 1992; Broadgate et al., 1997). Chlorophyll-a has been widely employed as a measure of phytoplankton
abundance and a proxy for predicting isoprene concentrations in water (Hackenberg et al., 2017). The
satellite-derived chlorophyll-a level during the study period over the SCS was below 0.45 mg/m$^3$, excluding
coastal areas (Fig. S3). The observations of $11\pm14$ ng/m$^3$ in Category 2 should be considered as the upper
limitation value derived from marine phytoplankton in the SCS. Although air masses differed between
Categories 1 and 2, a good correlation was obtained between MTLs and 2-MGA when the data in the two
categories was pooled for analyses ($R^2 = 0.77$, $P < 0.01$). This strong correlation indicated these tracers are
primarily formed through shared pathways. However, this correlation was poor over the NWPO, as discussed
below.
3.6 Origin and formation of $SOA_I$ over the NWPO
Over the NWPO, the concentration of 2-MGA was $1.6\pm1.5$ ng/m,$^3$ which was generally dominant among $SOA_I$
tracers, followed by MTLs ($0.7\pm0.3$ ng/m$^3$) and C5-alkene triols ($0.03\pm0.02$ ng/m$^3$). When the ratio of
2-MGA/MTLs was further examined, it varied greatly from <0.1 to 6.3, with a median value of 2.1. Most ratios
observed over the NWPO in this study were far greater than the values of 0.18–0.59 reported by Hu et al. (2013a)
from a global circumnavigation cruise, and also greater than 0.87–1.8 reported in urban areas of California
(Lewandowski et al., 2013) and the maximum value of 2.0 obtained over the YBS. Ding et al. (2013) also
reported ratios that fluctuated greatly from 0.5 to 10 with a median value of 3.3 during a summer cruise in the
NWPO and Arctic Ocean in 2003. The large 2-MGA/MTL ratios over the NWPO appeared to be highly
consistent over two independent sampling campaigns.
The compound profile of $SOA_I$ tracers over the NWPO implied high-NOx conditions allowing oxidation of
isoprene to generate the $SOA_I$ present in most samples. Such high-NOx conditions are impossible in a remote
marine atmosphere, as indicted in Figure S2. Regarding the lifespan of isoprene in the atmosphere is only
several hours (Bonsang et al., 1992), the long-range transport of oxidation products formed under high $NO_x$
levels over the continents likely led to the 2-MGA-dominated composition of $SOA_I$. Based on air mass back
trajectories, this long-range transport may involve 2-MGA originating from Siberia, northeastern China, or
Japan.
Organic aerosols over the NWPO were strongly influenced by forest fires that take place in Siberia during
spring and summer almost every year (Ding et al., 2013; Huang et al., 2009). Previous emissions inventory
studies have reported high isoprene and $NO_x$ emissions from various BB types (Akagi et al., 2011; Andreae and
Merlet, 2001). Ding et al. (2013) thus argued that an increase in emissions of isoprene in the presence of BB,
followed by its chemical conversion under high-$NO_x$ conditions, may lead to transport over thousands of
kilometers and hold at the detectable concentrations in the remote marine atmosphere over the NWPO. The
same argument may hold true for the elevated ratios of 2-MGA/MTLs observed over the NWPO in this study
(Fig. 4). However, we did not find a significant correlation between 2-MGA and LEVO over the NWPO.
On the other hand, the ratios of 2-MGA/MTLs in 3 of 19 samples collected over the NWPO were below 0.5
(Figure 4). In these cases, the oxidation of isoprene under low-NOx conditions likely dominated the generation
of SOA$_I$. The ratios of 2-MGA/MTLs were 0.5–1.5 in 4 of 19 samples, suggesting mixed contributions to SOA$_I$
from the oxidation of isoprene under low-NOx conditions and high-NOx conditions. As the major formation
pathways of 2-MGA and MTLs varied greatly among samples, no significant correlation ($R^2 = 0.12$, $p > 0.05$)
was obtained between 2-MGA and MTLs over the NWPO. Recall that the tracer values of SOA$_I$ were 2.7±1.8
ng/m$^3$ in Category 1 and 1.7±1.0 ng/m$^3$ in Category 2. This implied that SOA$_I$ derived from marine sources was
comparable to that derived from the continent outflows.

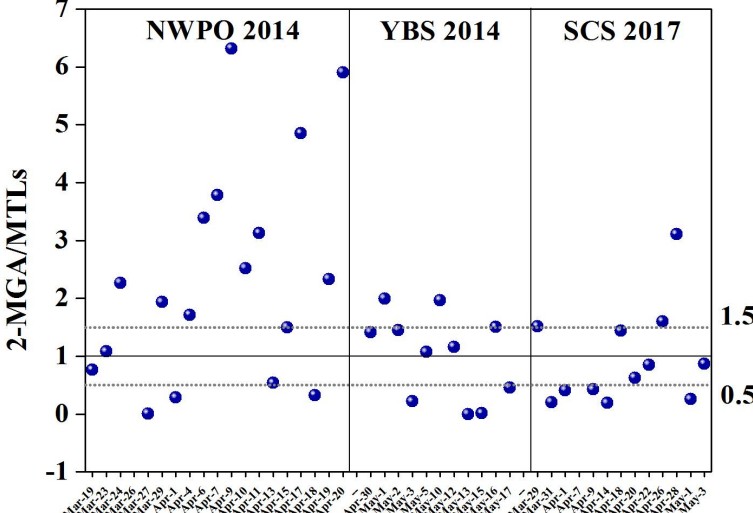


**Figure 4. Spatial ratio of 2-MGA/MTLs among SOA$_I$ tracers over three marine regions.**
3.7 Source apportionment of secondary organic carbon (SOC)
The tracer-based approach developed by Kleindienst et al. (2007) was applied to estimate the concentrations of
SOC and WSOC$_{BB}$, as follows:

$$[SOC] = \frac{\sum_i[tri]}{f_{SOC}}$$

376 (1)

$$[WSOC_{BB}] = \frac{C_{tracer}}{f_{tracer/WSOC_{BB}}}$$

377 (2)

where $\sum_i(tri)$ is the sum of concentrations of the selected suite of tracers for a precursor, and f$_{SOC}$ is the mass
fraction of tracer compounds in SOC generated from the precursor in chamber experiments. Assuming that the
f$_{SOC}$ values in ambient air match those in the chamber, the f$_{SOC}$ values for precursors such as isoprene,
monoterpenes, β-caryophyllene, and aromatics were 0.155±0.039 µg/µgC, 0.231 ± 0.111 µg/µgC, 0.023±0.0046





μg/μgC, and 0.00797 ± 0.0026 μg/μgC, respectively (Kleindienst et al., 2007), with uncertainty described in
Sect. S2. The fraction of LEVO in WSOC (0.0994 μg/μgC) from the BB plume was used for $WSOC_{BB}$ (Ding et
al., 2008). The $f_{SOC}$ value for monoterpenes was scaled up by a factor of 3.1 based on experimental observations,
as these two tracers (HGA+HD-MGA) accounted for 2/9 of the total tracers of monoterpenes, as described in
the supporting information (Kleindienst et al., 2007).
Over the SCS, nearly half of the sum of SOC and $WSOC_{BB}$ was in the form of $SOC_I$ (47%), followed by $SOC_A$
(36%), $WSOC_{BB}$ (14%) and a minor contribution of 2.5% from $SOC_M$ (Fig. 5). This composition pattern over
the SCS could be attributed to abundant biogenic SOA formation in low-latitude tropical marine atmospheres.
Over tropical marine regions, atmospheric oxidation products can account for 47–59% of the total organic
content estimated, with biomass burning emissions making up only 2–7% based on source apportionment using
organic tracers (Fu et al., 2011). A model study by Fu et al. (2012) showed that secondary formation accounts
for as much as 62% of OC estimated using tracers in eastern China in summer. A reverse pattern was observed
over the YBS, with $WSOC_{BB}$ as the dominant contributor (45%) to the sum of SOC and $WSOC_{BB}$, followed by
$SOC_A$ (32%) and $SOC_I$ (20%). The contribution of $SOC_M$ was also minor, at 1.5%. Notably, the chemical
composition observed over the NWPO was similar to that over the YBS, with $WSOC_{BB}$ contributing up to 53%.
In addition, Kang et al. (2018) used the PMF method to identify various sources of OC in marine aerosols over
the ECS such as secondary nitrate, BSOA, BB, and fungal spores.
Geographically, the estimated SOC values from BVOCs ranked at the highest level of 306±343 $ngC/m^3$ over the
SCS, decreasing to 107±99 $ngC/m^3$ over the YBS and 24±22 $ngC/m^3$ over the NWPO. The estimates of
aromatic SOC exhibited the same geographic trend, with values of 225±208 $ngC/m^3$ over the SCS, 151±177
$ngC/m^3$ over the YBS and 48±69 $ngC/m^3$ over the NWPO. Recent modeling results have also shown that
aromatic emissions are the predominant precursors of SOA during springtime in China in comparison with
BVOCs and other AVOCs (Han et al., 2016). Among estimates of $WSOC_{BB}$, the highest values of 209±108
$ngC/m^3$ were recorded over the YBS, followed by comparable levels of 86±98 $ngC/m^3$ (SCS) and 83±145
$ngC/m^3$ (NWPO).
In our study, the calculated $WSOC_{BB}$ estimate accounted for 4.1±5.0% and 3.3 ±1.7% of measured OC over the
NWPO and YBS, respectively, and these values are higher than that obtained over the ECS during summer
(1.4%) (Kang et al., 2018). Estimated SOC from BVOCs accounted for only 1.5±1.4% and 1.8 ±1.7% to the
measured OC over the NWPO and YBS, respectively, which is lower than that over ECS (4.21%) (Kang et al.,
2018). However, the mean values obtained in this study were similar to the total SOC level estimated using
tracers as a proportion of measured WSOC (4%) during a cruise on the North Pacific and Arctic Oceans,
supposed that WSOC accounted for half of the total OC in atmospheric particles (Ding et al., 2013).
The calculated SOC level derived from organic tracers accounted for less than 6% of total measured OC in these
study areas. However, this SOC compounds are expected to derive mainly from photochemical reactions in the
gas phase, followed by gas-aerosol partitioning. These compounds likely play an important role in the growth of
newly formed particles alongside pre-existing nucleation mode or Aitken mode particles. However, most organic
matter detected in bulk samples may originate from primary sources, heterogonous reactions and in-cloud
processing (Ervens et al., 2011; Kanakidou et al., 2005; Nichols, 2016), and these compounds may be major
drivers of the direct climate effects of aerosols, rather than indirect climate effects. In the future, a
comprehensive combination measurement of organic tracers and organic matter with an aerosol mass
spectrometer should be used to elucidate the formation and growth processes of atmospheric nanoparticles.



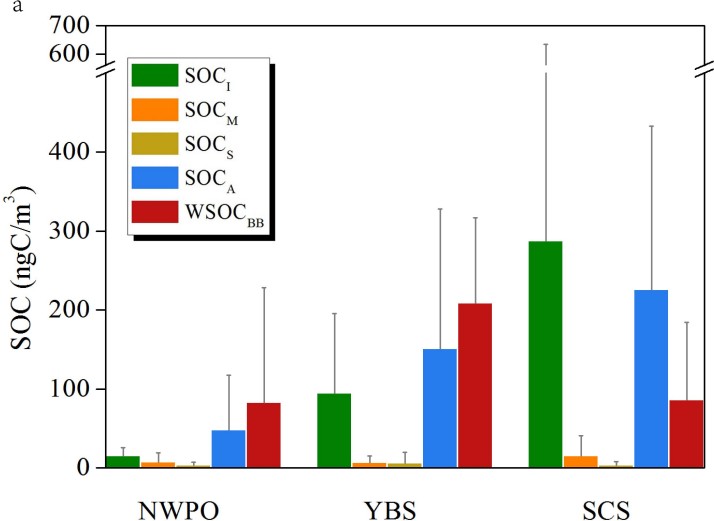

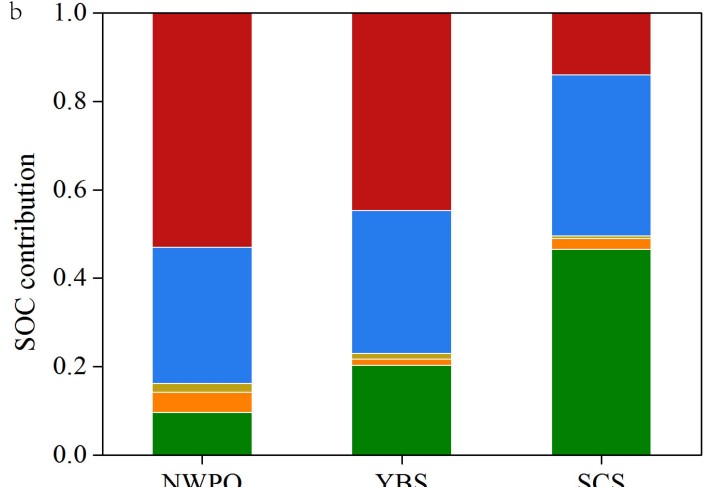


**Figure 5. Average SOC levels calculated using the tracer-SOC/WSOC method over three marine regions**
**(ECS and NWPO in 2014, SCS in 2017) and their contributions based on five organic tracers measured in**
**this study.**
**4. Conclusions**
This study investigated the geographical distributions of tracer-based organic matter observations in TSP
collected over two marginal seas of China and the NWPO in the spring season, when the East Asian monsoon
carries biogenic and anthropogenic aerosols over these oceanic zones. We found that a significantly large
difference in LEVO over the NWPO between two categories of air masses originating from upwind continents
or oceanic regions, as Category 1 (continental) contained 13±18 ng/m³ and Category 2 (oceanic) had 2.0 ±1.8





ng/m³; the concentrations of LEVO in Category 2 were closer to the low values reported in the literature. This
further implied a large increase in contribution of continent-derived BB aerosols to marine atmospheres over the
NWPO in recent decades, compared to previous studies. Combining the L/M ratios of 19±4 over the NWPO
with the calculated air mass back trajectories indicates that the increase was very likely associated with
enhanced emissions of BB aerosols from wildfires in Siberia and northeastern China. Moreover, the mean level
of BB aerosols over the SCS nearly matched that over the NWPO. The contents of LEVO in Category 2 air
masses, derived from oceanic zones over the SCS, were comparable with those reported in the literature, but the
mean value was only about a quarter of that in Category 1, representing air masses from upwind continents.
However, the limited data available over the SCS in the literature cannot support inferences about whether BB
aerosols emitted from upwind tropical forests have increased in recent decades.
The concentrations of SOA$_I$ over the SCS were approximately one order of magnitude greater than those
observed over the NWPO and several times larger than those over the YBS. The larger values observed over the
SCS in Category 1 than in Category 2 were likely driven by high emissions of isoprene from upwind tropical
forests and strong solar radiation. The MTLs dominance of SOA$_I$ over the SCS strongly suggested that SOC
from BVOCs was generated primarily under low-NO$_x$ conditions. On the other hand, 2-MGA dominance over
the YBS implied that most SOC was generated under high-NO$_x$ conditions. Elevated ratios of 2-MGA/MTLs
of >1.5 were obtained for 11 of 19 total samples collected over the NWPO, consistent with those reported in the
literature. Larger ratios may be attributed to possible emissions of BVOCs in the presence of BB. However, the
comparable concentrations of SOA$_I$ in Category 1 and Category 2 samples collected over the NWPO implied a
large contribution of SOA$_I$ from marine sources. The aromatic SOA tracers' levels were highest over the SCS,
followed by values obtained over the YBS and NWPO. The high values observed over the SCS may be related
to strong solar radiation, but the sources of precursors remain unexplained. Based on the concentrations in
Category 1 and 2 air samples collected over the SCS and NWPO, mixed sources of aromatic VOCs should exist,
including continent-derived precursors, oil exploration and heavy marine traffic.
Over the NWPO and the YBS, the estimated WSOC$_{BB}$ levels were nearly equal to the sum of SOC estimated
from the oxidation of aromatics and BVOCs. Over the SCS, SOC estimated from the oxidation of BVOCs was
significantly larger than the estimated WSOC$_{BB}$. The geographical difference may be related to emissions of
primary particulate organics and gaseous precursors as well as formation processing of secondary organics in
various atmospheres.
The atmospheric composition of SOA in different geographical locations is, however, highly complex and is
regulated by many factors including local meteorological conditions, anthropogenic emissions, plant species,
vegetation cover and regional chemistry, and therefore warrants further quantification and analyses. Particularly,
whether BB aerosols and other biogenic organic aerosols in marine atmospheres will continuously increase
under warming conditions.



**Table 1. Sum of organic tracer contents (ng/m³) at different locations worldwide.**

| Site | Date | Sampler | LEVO | SOA$_I$ | SOA$_M$ | SOA$_S$ | SOA$_A$ | Reference |
|---|---|---|---|---|---|---|---|---|
| **Wakayama, Japan (Forest)** | August 20–30, 2010, Day | TSP | 2.5±2.1 | 281±274 | 54.6±50.2 | 1.2±1.2 | | (Zhu et al., 2016a) |
| | Night | | 1.1±0.9 | 199±207 | 36.3±33.6 | 0.9±0.8 | | |
| **Across China** | summer 2012 | Anderson sampler | | 123±79 | 10.5±6.6 | 5.0±4.0 | 2.9±1.5 | (Ding et al., 2014) |
| **Beijing (PKU) (urban site)** | summer 2007 | PM2.5 | 37-148 | 59±32 | 30±14 | 2.7±1.0 | | (Yang et al., 2016) |
| **Beijing (YUFA) (suburban site)** | | | 34-149 | 75±43 | 32±14 | 3.9±1.5 | | |
| **Shanghai (BS) (Suburban site)** | Apr-May 2010 | PM2.5 | 88.8±57.2 | 3.8±3.9 | 6.1±3.7 | 1.0±0.7 | 1.1±0.7 | (Feng et al., 2013) |
| **Shanghai (XJH) (Urban site)** | | | 58.3±27.5 | 2.5±1.7 | 2.7±1.3 | 0.4±0.3 | 0.6±0.4 | |
| **Mt. Tai** | summer 2014 | PM2.5 | | 56.4±45.6 | 34.4±28.4 | | | (Zhu et al., 2017) |
| **Central Pearl River Delta** | fall-winter 2007 | PM2.5 | | 30.8±15.9 | 6.6±4.4 | 0.5±0.6 | | (Ding et al., 2011) |
| **Central Tibetan Plateau** | 2012-2013 | Anderson sampler | | 26.6±44.2 | 1.0±0.6 | 0.09±0.1 | 0.3±0.2 | (Shen et al., 2015) |
| **Mumbai, India** | winter 2007 | PM10 | | 4.1±2.4 | 29±22 | | 0.6±0.6 | (Fu et al., 2016) |
| | summer 2007 | | | 1.1±0.7 | 9.4±4.7 | | 0.05±0.1 | |
| **Alaska** | Spring 2009 | TSP | | 2.4 | 3.6 | 0.9 | | (Haque et al., 2016) |
| | 2008-2009 | TSP | | 4.1 | 2.0 | 1.5 | | |
| **SYS** | Spring 2017 | TSP | 9.6±8.6 | 45±54 | 3.5±6.0 | 0.07±0.1 | 1.8±1.7 | This study |
| **YBS** | Spring 2014 | TSP | 21±11 | 15±16 | 1.6±2.0 | 0.1±0.3 | 1.1±1.4 | This study |
| **NWPO** | Spring 2014 | TSP | 8.2±14 | 2.3±1.6 | 1.6±2.7 | 0.05±0.09 | 0.3±0.5 | This study |
| **East China Sea** | 18 May to 12 June 2014 | TSP | 0.09–64.3 (7.3) | 0.15–64.0 (8.4) | 0.26–87.2 (11.6) | 0.16–17.2 (2.9) | | (Kang et al., 2018) |
| **Arctic to Antarctic** | July to September 2008; November 2009 to April 2010 | TSP | 5.4±6.2 | 8.5±11 | 3.0±5.0 | | | (Hu et al., 2013a; Hu et al., 2013b) |
| **North Pacific** | 2003 | TSP | | 0.5±0.4 | 0.6±0.4 | 0.06±0.05 | 0.002±0.005 | (Ding et al., 2013) |



**Ocean and the
Arctic**



**Data availability.** Most of the data are shown in supplement. Other data are available by contacting the
corresponding author.
**Supplement.** The supplement related to this article is available.
**Author contributions.** XY, TG and JF conceived and led the studies. TG, JW and JF carried out the
experiments and analyzed the data. TG and JF interpreted the results. ZG, JF, HG discussed the results and
commented on the manuscript. TG prepared the manuscript with contributions from all the co-authors.
**Competing interests.** The authors declare that they have no conflict of interest.
**Acknowledgements.** This research has been supported by the National Key Research and Development
Program in China (No.2016YFC0200504) and the Natural Science Foundation of China (Grant No. 41776086).

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
