# Peer review of "atmospheres from marginal seas of China to the northwest"

_Atmospheric Chemistry and Physics, 2019_

## Referee Comment (RC1) · Anonymous Referee #2 · 28 Nov 2019

The manuscript reports the spatiotemporal distributions of organic tracers in TSP collected in two marginal seas of China and the NWPO in the spring season and how the East Asian monsoon carries biogenic and anthropogenic aerosols over these oceanic zones. In addition, the authors discussed the origins of SOAI over the SCS and NWPO. Overall, it is an interesting and inspiring work. However, the follow comments need to be addressed before it can be accepted for publication on ACP.

Major comments: 1) Line 152-154: it's better to see if the levo/TSP ratio had been increased. otherwise it's inconclusive to say the contribution of BB aerosols to particle loading over the NWPO may have increased... 2) Line 227: the relative contribution of

[Figure]

SOA tracers to TSP in category 2 is much larger than that in category 1. Based on the authors' reasoning, is it realistic to infer that marine sources can contribute around 10% of TSP? 3) Line 294: what are the possible major precursors for DHOPA other than BB emission? 4) Line 362: it might be attributable to the different stability of 2-MGA and LEVO?

Minor comments: 1) Line 28: change "discuss" to "discussed" 2) Line 237: there is a redundant "burning" 3) Line 181: change "surprised" to "surprising" 4) Line 350: better to change "regarding" to "given that"

——————————————————————

---

## Referee Comment (RC2) · Anonymous Referee #3 · 28 Nov 2019

The manuscript of Guo et al. focuses on quantification of primary and secondary organic tracers in total suspended particles over the Yellow and Bohai seas (YBS) and the South China Sea (SCS) collected both in 2014, and the northwest Pacific Ocean (NWPO) collected in 2017. The authors focused on the long-range transport impact to the marine atmosphere in spring during East Asian monsoon season. Thus, the influence of continental BB aerosols in marine atmosphere was studied and the contributions of different precursors to the observed SOA were quantified using specific tracers. Also SOA formation pathways in marine atmosphere have been discussed and related to the literature data. It is high quality work focusing on organic aerosols from remote marine areas which are rare compared with continental ones.

[Figure]

Major comment: Major comment is related to the fact that this study presents a snapshot situation at particular marine areas and it is hard to distinguish how representative they are for general conclusions. With that in mind, the discussion and conclusions related to the SOA change in recent decades are questionable. This is especially addressed to the continent-derived BB aerosols affected by accidental intensive open-fire events and or specific meteorological conditions. Long-term study is needed to enable conclusions on increase/decrease in contribution of continental-derived BB aerosols to remote marine areas (L17-19; L152-154; L434-443). Despite the authors did great effort to support their discussion with the relevant literature, conclusions arising from there should be done more carefully since the authors compare a snapshot situation. Authors should comment on that and/or all general conclusions related to above should be avoided.

Minor comments

Line 153 Add . . .as a reference (Table 1),.. L161-L163 Remove –samples- from the sentence or rewrite L181 Change –surprised- with to surprise or surprising L205 I suggest to rewrite the first sentence in figure caption, it is unclearly written L212 Replace NWPO with SCS

---

## Author Comment (AC1) · 19 Jan 2020

Anonymous Referee #2 The manuscript reports the spatiotemporal distributions of organic tracers in TSP collected in two marginal seas of China and the NWPO in the spring season and how the East Asian monsoon carries biogenic and anthropogenic aerosols over these oceanic zones. In addition, the authors discussed the origins of SOAI over the SCS and NWPO. Overall, it is an interesting and inspiring work. However, the follow comments need to be addressed before it can be accepted for publication on ACP.

Response: We thank the reviewer's comments and revise our manuscript accordingly.

[Figure]

Major comments: 1) Line 152-154: it's better to see if the levo/TSP ratio had been increased. Otherwise it's inconclusive to say the contribution of BB aerosols to particle loading over the NWPO may have increased...

Response: The origin sentence is misleading and has been revised "Using these previous observations as a reference (Table 1), our observations suggested that the BB aerosols from the long-range transport over the NWPO in 2014 largely increased. Thus, an important question is raised, i.e., does the increase occur continuously and largely over the last decades in marine atmospheres over the NWPO?". The LEVO/TSP was 0.02% $\pm$ 0.03% (average $\pm$ standard deviation) and 0.02% $\pm$ 0.01% over the NWPO and over the YBS. It is meaningless to say the contribution of BB aerosols to particle loading over the NWPO.

2) Line 227: the relative contribution of SOA tracers to TSP in category 2 is much larger than that in category 1. Based on the authors' reasoning, is it realistic to infer that marine sources can contribute around 10% of TSP?

Response: We carefully check through the whole manuscript. We are sorry for the misleading, but we cannot find where cause this. In revision, we added "The average contribution of SOA tracers to TSP over the SYS was higher in category 1 (0.4% $\pm$ 0.6%) than in category 2 (0.06% $\pm$ 0.07%)." And "The average contribution of SOA tracers to TSP over the NWPO was higher in category 1 (0.008% $\pm$ 0.005%) than that in category 2 (0.005% $\pm$ 0.005%)."

3) Line 294: what are the possible major precursors for DHOPA other than BB emission? Response: The sentence has been revised as "leaving emissions other than BB emissions, e.g., solvent use, oil exploration, marine traffic, etc., as the major precursors for DHOPA in these marine atmospheres".

4) Line 362: it might be attributable to the different stability of 2-MGA and LEVO?

Response: Thanks for the suggestion. In revision, we added "The decomposition of

[Figure]

LEVO reported in literature (Hennigan et al., 2010; Hoffmann et al., 2010; Fraser and Lakshmanan, 2000) may lower the correlation between them. However, whether 2-MGA can decompose in ambient air remains poorly understood."

Reference: Fraser, M. P. and Lakshmanan, K.: Using levoglucosan as a molecular marker for the long-range transport of biomass combustion aerosols, Environ. Sci. Technol., 34, 4560-4564, https://doi.org/10.1021/es991229l, 2000.

Hennigan, C. J., Sullivan, A. P., Collett Jr., J. L. and Robinson, A. L.: Levoglucosan stability in biomass burning particles exposed to hydroxyl radicals, Geophys. Res. Lett., 37, L09806, https://doi.org/10.1029/2010GL043088, 2010.

Hoffmann, D., Tilgner, A., Iinuma, Y. and Herrmann, H.: Atmospheric stability of levoglucosan: A detailed laboratory and modeling study, Environ. Sci. Technol., 44, 694-699, https://doi.org/10.1021/es902476f, 2010.

Minor comments: 1)Line 28: change "discuss" to "discussed"

2) Line 237: there is a redundant "burning"

3) Line 181: change "surprised" to "surprising"

4) Line 350: better to change "regarding" to "given that"

Response: Done. Thanks for your advice.

Please also note the supplement to this comment: https://www.atmos-chem-phys-discuss.net/acp-2019-723/acp-2019-723-AC1-supplement.pdf
* * *

---

## Author Comment (AC2) · 19 Jan 2020

Anonymous Referee #3 The manuscript of Guo et al. focuses on quantification of primary and secondary organic tracers in total suspended particles over the Yellow and Bohai seas (YBS) and the South China Sea (SCS) collected both in 2014, and the northwest Pacific Ocean (NWPO) collected in 2017. The authors focused on the long-range transport impact to the marine atmosphere in spring during East Asian monsoon season. Thus, the influence of continental BB aerosols in marine atmosphere was studied and the contributions of different precursors to the observed SOA were quantified using specific tracers. Also, SOA formation pathways in marine atmosphere have

been discussed and related to the literature data. It is high quality work focusing on organic aerosols from remote marine areas which are rare compared with continental ones.

Response: We thank the reviewer's comments and revise our manuscript accordingly.

Major comment: Major comment is related to the fact that this study presents a snapshot situation at particular marine areas and it is hard to distinguish how representative they are for general conclusions. With that in mind, the discussion and conclusions related to the SOA change in recent decades are questionable. This is especially addressed to the continent-derived BB aerosols affected by accidental intensive open-fire events and or specific meteorological conditions. Long-term study is needed to enable conclusions on increase/decrease in contribution of continental-derived BB aerosols to remote marine areas (L17-19; L152-154; L434-443). Despite the authors did great effort to support their discussion with the relevant literature, conclusions arising from there should be done more carefully since the authors compare a snapshot situation. Authors should comment on that and/or all general conclusions related to above should be avoided.

Response: We agree with the comments and soften the arguments accordingly. In revision, they have been revised as "The comparison of levoglucosan observed in this study with values from the literature showed that the concentrations of biomass burning aerosols over the NWPO increased largely in 2014. More observations together with the snapshot measurement, however, need to confirm whether the large increase occurred continuously through the last decades."

"Using these previous observations as a reference (Table 1), our observations suggested that the BB aerosols from the long-range transport over the NWPO in 2014 largely increased. Thus, an important question is raised, i.e., does the increase occur continuously and largely over the last decades in marine atmospheres over the NWPO?"

"This further implied a large increase in continent-derived BB aerosols in marine atmospheres over the NWPO recently, compared to previous studies. An important question is thereby raised, i.e., does a large increase in continent-derived BB aerosols in marine atmospheres over the NWPO occur continuously and largely in recent decades?"

Minor comments

Line 153 Add . . .as a reference (Table 1),

L161-L163 Remove –samples- from the sentence or rewrite

L181 Change –surprised- with to surprise or surprising

Response: Done. Thanks.

L205 I suggest to rewrite the first sentence in figure caption, it is unclearly written

Response: The caption was rewritten into "Spatial distribution of LEVO in TSP over the NWPO in spring of 2014 and 72-hrs back trajectory associated with each TSP sample. The red lines represent that air masses can be derived from the continent (a, Category 1); the blue lines represent that air masses may be derived mainly from the oceans (b, Category 2). The red dots represent the locations of fires from Fire Information for Resource Management System (FIRMS, https://firms.modaps.eosdis.nasa.gov/). And the base map was from Resource and Environment Data Cloud 210 Platform, DOI: 10.12078/2018110201.".

L212 Replace NWPO with SCS.

Response: Done. Thanks.

Please also note the supplement to this comment:
https://www.atmos-chem-phys-discuss.net/acp-2019-723/acp-2019-723-AC2-supplement.pdf

---

## Author Response (AR2)

**Response to Editor**

Editor Decision: Publish subject to minor revisions (review by editor) (07 Feb 2020) by Willy Maenhaut

Comments to the Author:

The authors have reasonably well addressed the comments of the two anonymous referees and they have modified their manuscript accordingly. However, the comments given below should be addressed and several alterations are needed for the Main text and Supplement before the manuscript can be published in ACP.

**Response**:We thank the comments and revise our manuscript accordingly.

For the Main text:

Line 19: Replace "the snapshot" by "our snapshot".

**Response**:Done.

Line 20: Replace "to the mean" by "to a mean".

**Response**:Done.

Line 21: Replace "closer to" by "close to".

**Response**:Done.

Line 29: Replace "discussed the" by "discuss the".

**Response**:Done.

Line 51-54: Although SOA from the photo-oxidation of isoprene produced by phytoplankton blooms has been proposed as a potentially important marine SOA source by Meskhidze and Nenes (2006), a study by Claeys et al. (Chemical characterisation of marine aerosol at Amsterdam Island during the austral summer of 2006-2007, J. Aerosol Sci. 41 (2010) 13-22) found no evidence for isoprene SOA at that remote site. On the other hand, Gantt, Meskhidze, and Kamykowski (A new physically-based quantification of marine isoprene and primary organic aerosol emissions, Atmos. Chem. Phys. 9 (2009) 4915-4927) state "Using a fixed 3% mass yield for the conversion of isoprene to SOA, our emission simulations show minor (<0.2%) contribution of marine isoprene to the total marine source of OC on a global scale. However, our model calculations also indicate that over the tropical oceanic regions (30◦S to 30◦N), marine isoprene SOA may contribute over 30% of the total monthly-averaged sub-micron OC fraction of marine aerosol. The estimated contribution of marine isoprene SOA to hourly-averaged sub-micron marine OC emission is even higher, approaching 50% over the vast regions of the oceans during the midday hours when isoprene emissions are highest". This is in contrast with the study by Arnold et al. (Evaluation of the global oceanic isoprene source and its impacts on marine organic carbon aerosol, Atmos. Chem. Phys. 9 (2009) 1253-1262), where it is stated "Inclusion of secondary organic aerosol (SOA) production from oceanic isoprene in the model with a 2% yield produces small contributions (0.01-1.4%) to observed organic carbon (OC) aerosol mass at three remote marine sites in the Northern and Southern Hemispheres. Based on these findings we suggest an insignificant role for isoprene in modulating remote marine aerosol abundances, giving further support to a recently postulated primary OC source in the remote marine atmosphere". Note also that the study of Claeys et al. (2004) does not deal with marine isoprene SOA, but instead with isoprene SOA from the Amazon Basin. Using "marine AND SOA AND isoprene" as topic on the Web of Science I receive 47 hits. Therefore, the sentence here should be rewritten and appropriate references should be given. It is in any case clear that isoprene emissions from the continents are much more important than those from the oceans, as appears from the paper by Guenther et al. (A global model of natural volatile organic compound emissions, J. Geophys. Res. 100 (1995) 8873-8892), so that the importance of marine isoprene SOA should be downplayed.

**Response**:Thanks. We revise this sentence into "Secondary organic aerosols (SOAs) arising from the oxidation of phytoplankton-derived isoprene have been argued to affect the chemical composition of marine atmospheric aerosols and consequently impact CCN loading and cloud droplet number concentrations (Ekström et al., 2009; Meskhidze and Nenes, 2006), but the importance of the marine isoprene-derived SOA is still debated (Arnold et al., 2009; Claeys et al., 2010; Gantt et al., 2009; Guenther et al., 1995). For example, Gantt et al. (2009) estimated the contribution of marine isoprene-derived SOA to the OC in marine atmospheric particles ranged from <0.2% on a global scale, but to as high as 50% (sub-micron OC) over the vast regions of the oceans during the midday hours when isoprene emissions are highest." .

For the reference "(Claeys et al., 2004)", it was revised in "However, emission fluxes and oxidation processes of BVOCs show great variation, depending on global warming and other factors such as regional landscape, other pollutants in the ambient air, etc. (Ait-Helal et al., 2014; Claeys et al., 2004; Hu and Yu, 2013; Peñuelas and Staudt, 2010)." in line 69-71.

Line 56: Replace "future warming climate in the future" by "future warming climate".

**Response**:Done.

Line 61: See what I wrote above about marine isoprene SOA; its importance should be downplayed.

**Response**:Thanks. We revise this sentence into "More importantly, BVOCs emitted from continental ecosystems and their oxidation products can significantly affect the atmosphere in remote marine areas through long-range transport (Ding et al., 2013; Fu et al., 2011; Hu et al.,

2013a; Kang et al., 2018; Kawamura et al., 2017).".

Line 66: Replace "emissions fluxes" by "emission fluxes".

**Response**:Done.

Line 68: Replace "air etc." by "air, etc.".

**Response**:Done.

Line 71: Replace "Sharma," by "Sharma et al.,".

**Response**:Done.

Line 74: I do not understand the use of "revere" here; should it perhaps be "reverse" instead of "revere"?

**Response**:Done. We revise it into "reverse".

Line 75: Replace "Update observations" by "Updated observations".

**Response**:Done.

Line 77: Replace "we analyzed the" by "we determined the".

**Response**:Done.

Line 119: Replace "run every" by "ran every".

**Response**:Done.

Line 132: Replace "under the controlled" by "under controlled".

**Response**:Done.

Line 142: Replace "the smaller difference among the means" by "the small difference among the mean".

**Response**:Done.

Line 156: Replace "suggested that" by "suggest that".

**Response**:Done.

Line 170: Replace "indicated a" by "indicate a".

**Response**:Done.

Line 175: Replace "traceries and" by "trajectories and".

**Response**:Done.

Line 197: Replace "differences were found" by "difference was found".

**Response:** Done.

Line 198: Replace "limited samples" by "limited number of samples".

**Response:** Done.

Line 199: Replace "emissions sources" by "emission sources".

**Response:** Done.

Lines 230 and 233: I presume that it should be "SOAI" instead of "SOA".

**Response:** Done.

Page 9, Figure 3, right panel: Replace "SOA Traces" by "SOA Tracers".

**Response:** Done. And the revised figure is shown below.

[Figure]

Lines

240, 260, 408, 418, 420 and 436: It is unclear what "ECS" denotes; I presume that it stands for "East China Sea". In any case, abbreviations and acronyms should be defined (written full-out) when first used. Also, I think that "ECS" in two of these cases (i.e., in lines 240 and

436) should be replaced by "YBS".

**Response:** Thanks.

The "ECS" in line 240 and 436 was revised as "YBS". And abbreviations for "ECS" is defined (written full-out, "East China Sea") when first used in line 263-264.

Line 248: Replace "atmospheres (Gordon" by "atmosphere (Gordon".

**Response:** Done.

Line 266: Replace "confirm" by "confirm this".

**Response:** Done.

Line 276: Replace "remains unexplained" by "remain unexplained".

**Response:** Done.

Line 289: Replace "decrease down" by "decreases down".

**Response**:Done.

Line 292: Replace "differences were observed" by "difference was observed".

**Response**:Done.

Line 300: Replace "other two" by "the other two".

**Response**:Done.

Line 301: "Li et al., 2013" is missing in the Reference list; there is "Li et al., 2014" in that list to which not is referred within the text.

**Response**:Thanks. We revise the "Li et al., 2013" into "Li et al., 2014" in line 304, and the

"Li et al., 2014" within the text is referred in the reference list.

Line 339: Replace "MTLs observations" by "MTLs observation".

**Response**:Done.

Line 342: Replace "indicated these" by "indicates these".

**Response**:Done.

Line 425: Replace "this SOC compounds" by "these SOC compounds".

**Response**:Done.

Line 428: Replace "heterogonous reactions" by "heterogeneous reactions".

**Response**:Done.

Line 441: Replace "found that a" by "found a".

**Response**:Done.

Lines 728-729: There is no reference made to "Zhu et al., 2016b" within the text.

**Response**:The reference for "Zhu et al., 2016b" within the text was in line 66-69 in the revised text "BVOCs consist primarily of isoprene, monoterpenes, sesquiterpenes, and their oxygenated hydrocarbons such as alcohols, aldehydes, and ketones (Ehn et al., 2014;

Guenther et al., 2006) and account for the majority of the global VOC inventory (Heald et al.,

2008; Zhu et al., 2016a, b).".

**For the Supplement:**

Page 2, lines 1, 2, and 11: Replace "analyzed" by "measured".

**Response**:Done.

Page 2, line 12: It is unclear what is meant by "both analyses".

**Response**:Thanks. We revise this into "both analyses in this study and Kleindienst et al.

(2007)".

Page 6, line 2: Replace "up panel" by "upper panel" and replace "below panel" by "lower panel".

**Response:** Done.

Page 7, line 2: Replace "up panel" by "upper panel".

**Response:** Done.

Page 7, line 3: Replace "below panel" by "lower panel".

**Response:** Done.

Page 8, line 2: Replace "analyzed" by "measured".

**Response:** Done.

Page 9, line 1: Replace "primary, secondary" by "primary and secondary".

**Response:** Done.

Page 9, line 2: Replace "gaactosan" by "galactosan".

**Response:** Done.

[revised manuscript text omitted]

---

## Author Response (AR3)

*Response to editor*

Dear Prof. Willy Maenhaut,

Thank very much for your help and edition.

Best regards,

Xiaohong

Prof. Xiaohong Yao (Ph.D)

Ocean University of China

*Tracer-based investigation of organic aerosols in marine atmospheres from marginal seas of China to the northwest Pacific Ocean*

*Tianfeng Guo, Zhigang Guo, Juntao Wang, Jialiang Feng, Huiwang Gao, and Xiaohong Yao*

*For the Main text:*

*Lines 56-57: Replace "estimated the contribution of marine isoprene-derived SOA to the OC in marine atmospheric particles ranged from <0.2% on a global scale, but to as high as 50% (sub-micron OC) over the vast" by "estimated that the contribution of marine isoprene-derived SOA to the OC in marine atmospheric particles is <0.2% on a global scale, but that the hourly-averaged sub-micron OC emission may approach 50% over vast".*

*Line 108: Replace "analyzed in" by "measured in".*

*Line 132: Replace "were analyzed" by "were measured".*

**Response:** We have corrected these accordingly.

*For the Supplement:*

*Page 2, line 1: Replace "analyzed in" by "measured in".*

*Page 2, line 11: Replace "were analyzed" by "were measured".*

*Page 2, line 14: Replace "was analyzed" by "was measured".*

**Response:** We have corrected these accordingly.

*Furthermore, I noticed that your figures 1, 2, and 3 contain maps. To clarify the copyright, we differentiate between (a) maps entirely created by you, (b) maps created by you but based on layers reused from other originators, or (c) maps simply reused from other originators.*

*An example for (a) is a digital elevation model (DEM) purely based on measurement points collected by you and derived by using a software product. If you use an existing map layer from another originator as a basis for significantly enriching the map with your own content, this would be an example for case (b). Case (c) could be a pure reproduction of Google Maps where your own contribution is rather small (e.g. a city map where you only added a few marks for your study locations).*

*If the map was entirely created by you (case a), there is no need to change the caption or figure. However, if the map was not entirely created by you (cases b or c), please provide a new file in which the copyright is denoted in the figure itself. If this is not possible, please provide it in the caption.*

*Please make sure that the figure or caption contains the appropriate copyright statement as this is the responsibility of the authors.*

*Please let us know which case (a, b, or c) corresponds to the map used in your manuscript. Cases (b) and (c) need a copyright statement which consists of the copyright symbol © and the copyright holder (e.g. © Microsoft).*

**Response:** Thanks. The maps in figure 1, 2, 3 in this manuscript correspond to case b. And the copyright statement has been added both in the figures and the figure captions as shown below.

[revised manuscript text omitted]